# Distinct representations of basic taste qualities in human gustatory cortex

Junichi Chikazoe[1,2], Daniel H. Lee[3], Nikolaus Kriegeskorte[4] & Adam K. Anderson[2]

The mammalian tongue contains gustatory receptors tuned to basic taste types, providing an evolutionarily old hedonic compass for what and what not to ingest. Although representation of these distinct taste types is a defining feature of primary gustatory cortex in other animals, their identification has remained elusive in humans, leaving the demarcation of human gustatory cortex unclear. Here we used distributed multivoxel activity patterns to identify regions with patterns of activity differentially sensitive to sweet, salty, bitter, and sour taste qualities. These were found in the insula and overlying operculum, with regions in the anterior and middle insula discriminating all tastes and representing their combinatorial coding. These findings replicated at super-high 7 T field strength using different compounds of sweet and bitter taste types, suggesting taste sensation specificity rather than chemical or receptor specificity. Our results provide evidence of the human gustatory cortex in the insula.

[1] Section of Brain Function Information, Supportive Center for Brain Research, National Institute for Physiological Sciences, Aichi 4448585, Japan.
[2] Department of Human Development, Cornell University, Ithaca, New York 14850, USA. [3] Integrative Physiology, University of Colorado, Boulder, Colorado 80309, USA. [4] Department of Psychology, Columbia University, New York, New York 10027, USA. Correspondence and requests for materials should be addressed to J.C. (email: j.chikazoe@gmail.com) or to A.K.A. (email: aka47@cornell.edu)

Distinguishing safe from harmful food is critical for the survival of all organisms. So much so, unlike the distal senses of seeing and hearing, taste receptors are specifically tuned for discrete taste types associated with canonical hedonic values, serving as primary punishers and reinforcers[1]. For instance, sweet receptors aid in selection of energy-rich nutrients and bitter receptors guard against the intake of the potentially noxious, serving as the basis for oral distaste and disgust[2]. Although these taste receptors are distributed throughout the sensory surface of the oral cavity, rodent imaging studies have shown that these receptor types come together to form distinct regions in primary gustatory cortex in the insula, forming a potential gustotopic map of basic taste types[3, 4]. Our understanding of human gustatory cortex and its relation to taste experience, however, is much less developed, with no evidence discriminating basic tastes. In contrast, human cortical maps have characterized in detail other sensory systems, such as retinotopy in vision[5], tonotopy in audition[6, 7], and somatotopy in touch[8]. Here we employed multivoxel distributed activity of functional magnetic resonance imaging (fMRI) at 3 T and super-high field strength 7 T to map distinct taste qualities in the human brain. We examined whether discrete and/or distributed regions support specific taste qualities toward identifying the human gustatory cortex.

Beyond objective sensory qualities, basic tastes have strong subjective hedonic properties that are even present at birth[9]. In rodents, taste-type representations in primary gustatory cortex appear to have built-in appetitive or aversive responses[10] such that objective taste types are associated with their palatable value (e.g., sweet is pleasant, bitter is unpleasant). Unlike the distal senses, hedonic attributes may be a primary dimension of the chemical senses[11], present in primary gustatory cortex. Further, we may encode more abstract valence representations across sensory systems[12], such as the beauty derived from the photons of a landscape in sunset or the pleasure of a pinot noir atop its bitter molecular composition. The challenge of identifying taste-specific responses in humans may be related to the complexity of our taste experiences being associated with these multiple levels of objective chemical and subjective valence. Indeed, human fMRI has shown that putative primary gustatory cortex in the insula is responsive to at least two associated levels of representation: taste qualities and their palatability[12, 13].

The human primary gustatory cortex is presumed to lie in the insular cortex[14]. However, tastes evoke activity in multiple regions in the human brain, including the insula, frontal operculum, parietal operculum, and orbitofrontal cortex[13, 15]. These regions also tend to show similar responses to different taste types[14], with any observed differences reflecting hedonic experience[12] rather than differences in taste quality. Although representations of hedonic value are found in many regions in the brain[13], classification of basic tastes and thus relative taste specificity should be the defining feature of gustatory cortex and its boundaries in the human brain.

To define human gustatory cortex, we used multivoxel activity patterns to assess where basic taste representations reside and how they are organized. Prior fMRI studies of taste have employed univariate analyses, where each voxel response is analyzed separately. By contrast, multivoxel pattern analysis of fMRI activity analyzes fine-grained patterns of activity by combining evidence across voxels. It has been used to reveal representations that are distributed across[16] and within brain regions[17], such as fine-grained representations of orientation tuning in primary visual cortex[18, 19]. To test whether localized pattern activity supports a gustotopic map in the human brain, we employed a multivoxel pattern analysis using a "searchlight" method[20]. Within a searchlight sphere (radius: 4 mm), we

examined the discriminability of taste types, enabling us to localize regions distinguishing different tastes by their activity patterns, how they are organized, and to what degree they can be dissociated from palatability, the hedonic component of taste.

## Results

**Univariate analysis of voxel-specific taste tuning**. We first conducted an fMRI experiment at 3 T in which participants ingested four basic tastes (i.e., sour, sweet, bitter, and salty, pre-matched for intensity within each participant) and one tasteless artificial saliva solution. After each taste trial, participants rated the experience along two independent sliding scales for positive (pleasant) and negative (unpleasant) hedonic valence, allowing a more detailed analysis of palatability, where a taste can combine pleasant and unpleasant components[21]. Despite clear average differences across tastes, ratings of hedonic valence also varied within a participant across the experiment (Supplementary Table 1). This trial-by-trial hedonic variation afforded separate modeling of subjective palatability and objective taste type. Although the source of trial-by-trial variability is unknown, it did not reflect hedonic contrast (e.g., bitter following sweet or sweet following bitter), which did not affect valence ratings, likely due to the long stimulus onset asynchrony ($ps > 0.16$).

Electrophysiological[22] and anatomical tract tracing studies[23] in nonhuman primates have identified the dorsal anterior insula as gustatory cortex[24]. We thus began our examination by outlining a structural region of interest encompassing the entire insular cortex (Fig. 1a, white outline). We first examined whether basic tastes are coded by segregated regions in the insula via traditional univariate measures that consider voxel-specific tuning. The results showed a variety of voxel-specific taste activations within individuals, which were inconsistent across individuals for the same taste (Fig. 1). In aggregate, these results were similar to previous work[25] showing little evidence of activity to individual tastes at the group level (Fig. 1a, Supplementary Figure 1). This did not reflect a lack of sensitivity to individual tastes, as we did find clear discrimination of bitter taste relative to resting baseline that persisted when compared against tasteless solution ingestion (Supplementary Figure 1), but was abolished when controlling for ratings of hedonic valence (thresholded at familywise error rate (FWE) < 5%, small volume correction). No voxel achieved significance when contrasted against each other (e.g., sour vs. bitter). As an important confirmation of prior findings[15], univariate analyses not controlling for valence revealed consistent neural correlates of bitter taste experience relative to tasteless in the whole brain, activating the insula, frontal operculum, parietal operculum, thalamus, and cingulate gyrus (Supplementary Figure 2 and Supplementary Table 2; conducted at a more liberal threshold false discovery rate (FDR) < 5%).

To test the existence of voxel-specific taste tuning, we split each participant's odd and even runs, comparing the voxel activity for each taste in the odd runs to the voxel activity for each taste in the even runs. For illustration, when voxels were rank-ordered based on activation to each taste in even runs, we found consistent patterns of correspondingly decreasing activation for all four tastes in odd runs (Fig. 1b). Computing the voxel correlations between all taste types showed no taste specificity across participants, i.e., a confusion matrix showing insular voxel activity to each taste corresponded to all other tastes (Fig. 1c, d). The lack of taste-specific responses renders the precise localization of human gustatory cortices uncertain, in particular when palatability is taken into account.

**Multivoxel pattern analysis of taste-specific tuning**. We next applied multivoxel pattern analyses incorporating information

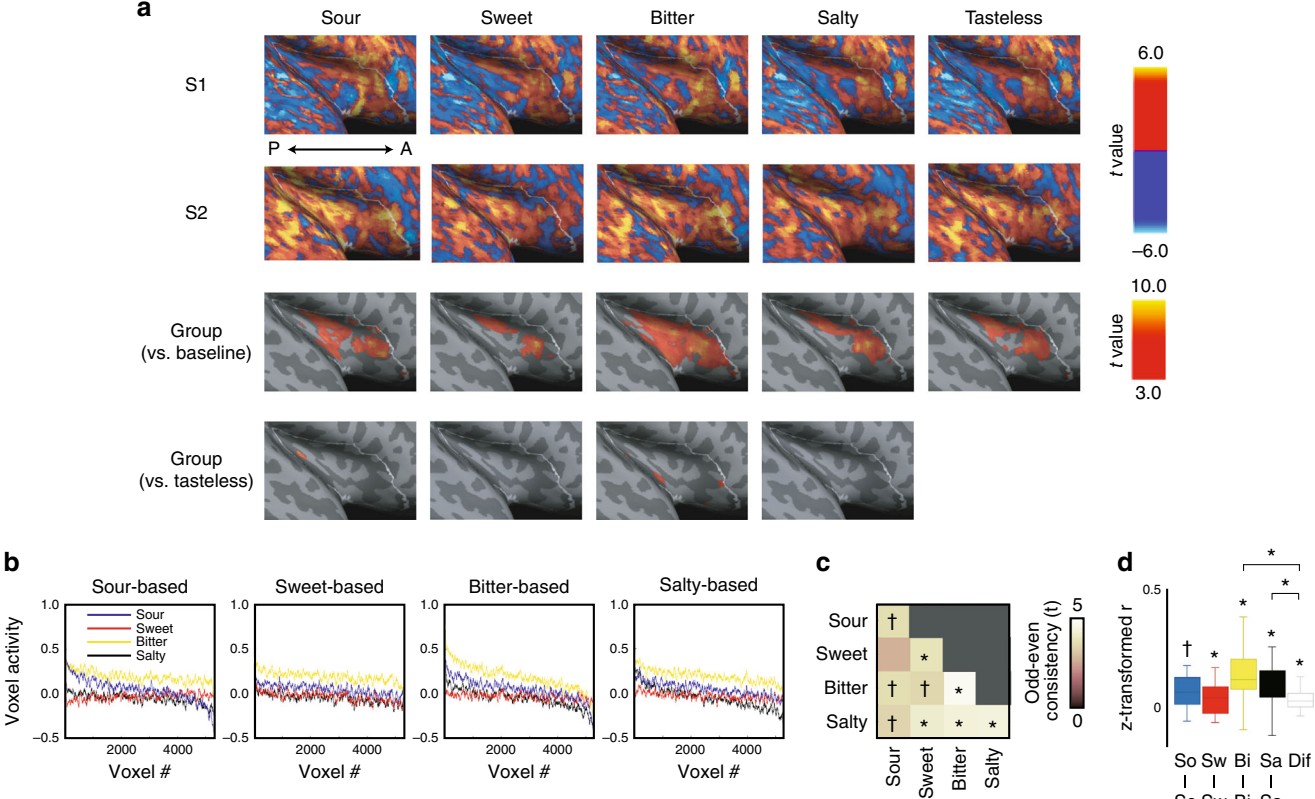

**Fig. 1** Univariate analysis of voxel-specific taste tuning in the insula. **a** Upper two rows indicate activation maps for sour, sweet, bitter, salty, and tasteless stimuli of two subjects S1 and S2. The third row indicates group results when contrasted against baseline fMRI activity ($n = 20$ participants). The fourth row indicates group results when contrasted against tasteless, to control for non-taste-related gustatory activity and swallowing. When estimated against tasteless, controlling for positive and negative hedonic values, no taste-sensitive voxels survived statistical significance. A: anterior, P: posterior. **b** Each participant's voxel activity in odd runs were aligned based on rank-ordered sensitivity to each sour, sweet, bitter, and salty taste in even runs, then averaged across participants. The corresponding downward trend for all tastes in each panel shows a lack of taste specificity. **c** Correlations were computed for voxel activation between odd and even runs between all taste combinations within each participant, submitted to one-sample $t$-test across participants. Corresponding activation between taste types shows a lack of taste-specific voxel tuning in the insula. **d** Correlations between odd and even runs for all same taste and different taste combinations within each participant. Correlation coefficients were $z$-transformed and subject to one-sample $t$-test across participants. Boxes represent the median and 25th/75th percentiles and whiskers represent the minimum and maximum. Bi: bitter, Dif: different, Sa: salty, So: sour, Sw: sweet. †$p < 0.05$ uncorrected, *$p < 0.05$ after Bonferroni correction for multiple comparisons (FWE < 5%)

from local patterns of activity expressed across multiple voxels. Using a linear discriminant analysis (LDA) classifier, a multivoxel searchlight[20] explored whether activation patterns in the insular cortex were capable of discriminating each basic taste type. Given evidence of taste specificity in nonhuman animals, we hypothesized that taste-type representations are spatially separated in the insula but may be best detected by multivoxel patterns. We employed a smaller radius (4 mm) for the searchlight to avoid underestimation of spatial separation between potential taste-specific regions. This approach produced four taste discriminability maps wherein the activation pattern for each taste type was discriminated from each other taste type, e.g., the "sweet" cluster supported pairwise discriminations from each bitter, salty, and sour; see Methods for statistical details) (Fig. 2a). By conjoining four basic taste discriminability maps, we found the anterior/middle insula differentially responded to four basic tastes (Fig. 2b), which is consistent with putative primary gustatory cortex based on anatomical projections from the gustatory thalamus in nonhuman primates[23].

To further examine how each taste type was coded relative to each other and its correspondence across people, we assessed the representational geometry of insular voxel activity patterns. First,

for each individual, we defined a region of interest (ROI) within the insula that discriminated all four tastes based on all other participants' data. To demonstrate the consistency across individuals, these resulting regions are illustrated in Fig. 2c with an indication of the proportion of significant maps. Using this ROI, a taste discriminability matrix was made for each subject. This leave-one-out procedure was repeated such that each subject is used once as the test data, ensuring independence of fMRI signal characterization from ROI definition. Classification performance in the right and left hemisphere demonstrated no significant difference after corrections for multiple comparisons and thus were collapsed. The result was a representational dissimilarity matrix of the average classification performance for each pair of taste discrimination (e.g., sour vs. sweet) across individuals, revealing that multivoxel patterns supported discrimination of all taste-type pairs (Fig. 2d).

Taste stimuli such as sucrose and quinine differ not only in their sweet and bitter quality, but also in hedonic value, i.e., palatability. Although prior work suggests there is a separation of hedonic features in the chemical senses[13, 26], multivariate patterns can reveal coding of information that is missed by traditional univariate approaches, such as hedonic valence in the insular

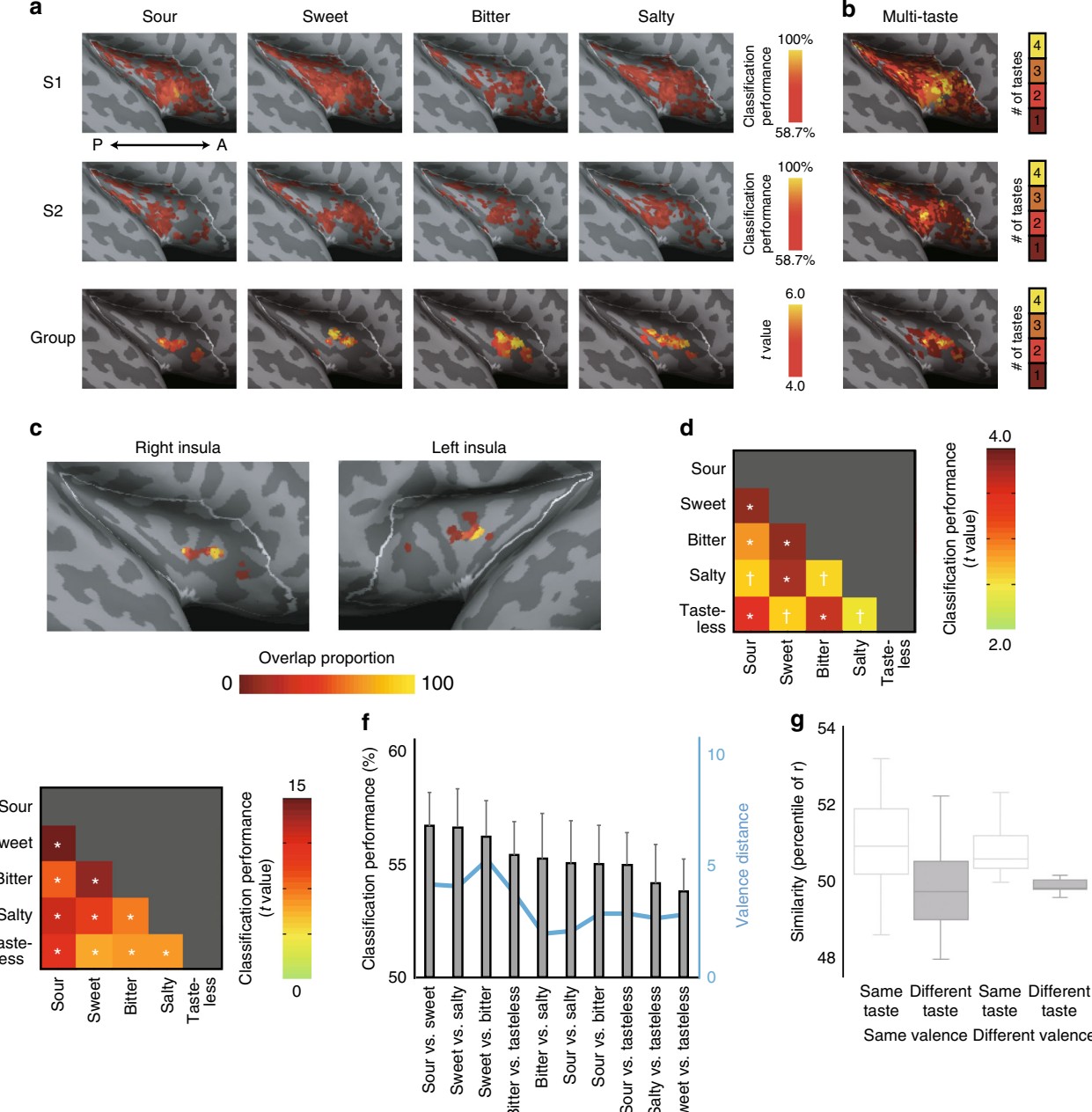

**Fig. 2** Multivoxel pattern analysis of taste-specific tuning in the insula. **a** Upper two rows indicate discriminability maps for sour, sweet, bitter, and salty stimuli of individual subjects (same as in Fig. 1a). Discriminability map indicates regions in which the average pairwise classification performance for a specific taste vs. each other taste was significantly higher than chance (50%). The linear discriminability searchlight maps (searchlight radius = 4 mm) were thresholded at $p < 0.05$ (uncorrected), which corresponded to 58.7% discrimination (chance level: 50%). The third row indicates group multi-taste-coding results ($n = 20$ participants) in which subjects were treated as a random effect, thresholded at FWE < 5% (small volume correction). A: anterior, P: posterior. **b** The number of taste types represented in the insula. Multi-taste population codes discriminated four basic tastes in the anterior and middle insula. **c** Overlap of four taste discriminability maps (each satisfies FWE < 5%) by leave-one-subject-out procedure. Color code indicates the percentage among the 20 leave-one-subject-out maps that was significant. **d** Representational dissimilarity matrix showing the pattern discriminability for each taste pair based on fMRI activity patterns of the right and left insula. **e** Representational dissimilarity matrix showing the pattern discriminability for each taste pair based on positive and negative valence ratings. **f** Classification performance for each taste pair compared with valence distance suggests discrimination of taste type separate from valence. Error bars indicate SE. **g** Repeated-measures ANOVA with taste type and valence as factors and trial fMRI activity pattern similarity (measured as correlations) as dependent variable. A main effect of taste type, but not valence nor interaction, demonstrates insular taste-type patterns do not reflect differences in experienced valence. Boxes represent the median and 25th/75th percentiles and whiskers represent the minimum and maximum. †$p < 0.05$ uncorrected, *$p < 0.05$ after Bonferroni correction for multiple comparisons (FWE < 5%)

cortex[12]. Thus, we explored whether the taste-type discrimination shown above is explained by an individual's subjective hedonic response. We first computed a representational dissimilarity matrix showing that objective taste types could be discriminated by their subjective valence ratings (Fig. 2e). We then show that pairwise discrimination of taste types in the insula did not relate to the degree of difference in valence across tastes (Fig. 2f). To statistically examine the separate contributions of taste type and valence in classification, we conducted an analysis on the four-taste discrimination maps derived above. We computed the similarity of insular activation pattern between trials, sorting them into two independent factors of taste type (same and different) and hedonic valence (same and different). We submitted these similarity data to a $2 \times 2$ repeated-measures analysis of variance (ANOVA), which revealed a main effect of taste type ($F_{(1,19)} = 19.5$, $p < 0.001$; Fig. 2g and Supplementary Figure 3) but no main effect of valence ($F_{(1,19)} = 0.7$, $p = 0.43$) and no interaction ($F_{(1,19)} = 0.4$, $p = 0.51$).

Although evidence from electrophysiology[22] and tract tracing[23] has focused on the subregions of the insula as putative gustatory cortex, gustatory discriminative regions may also reside in regions outside of the insula. Several previous studies[27, 28] argue that the primary taste region resides in the frontal/parietal operculum in addition to the middle insula[29]. To address taste discrimination using our multivoxel patterns discrimination approach, we expanded the searchlight analysis to the whole brain. Although there was evidence for responses to single tastes throughout the brain, coding of four basic taste types was regionally selective. In addition to insular subregions, the frontal and parietal operculum contained information about the four taste types (Fig. 3 and Supplementary Table 3). Thus, multivoxel patterns supporting multi-taste discrimination is not a property widely found throughout the brain. Its specificity provides a new functional definition of gustatory cortices as residing in the anterior/middle insula and extending to the anatomically associated[30] overlying fronto-parietal operculum.

**Specificity of taste quality under super-high field strength**. Selective responses to tastes can reflect chemical or receptor specificity rather than sensory quality. For instance, considering the variety of bitter taste receptors[31], the overlapping taste representations may be specific to quinine but not to other bitter tastes, thus reflecting the compound rather than the bitter sensory experience. We thus conducted a second experiment that employed two new variants of bitter (catechin and magnesium chloride) and sweet tastes (glucose and sucralose) to provide convergent and discriminant validation of multivoxel pattern characterizations of taste quality. To examine mapping of taste quality representations at higher voxel resolution, we conducted the experiment in super-high 7 T magnetic field.

We found bitter and sweet representations in the insular cortex. First, univariate analyses showed that, within individuals, insular voxels sensitive to one bitter or sweet taste were also sensitive to the other bitter or sweet taste but not across taste type (Fig. 4a, b and Supplementary Figure 4a), supporting taste-specific representations in the primary gustatory cortex, not chemical-specific representations. However, univariate analyses at the group level again showed no voxels that survived multiple comparisons for each taste vs. tasteless contrast. When insular voxels sensitive to taste for the group were tested for each subject left out, taste-type voxels were no longer consistently activated (Fig. 4c, d and Supplementary Figure 4b), reiterating the high inter-individual topographical variability of taste representations[25]. Lastly, we applied multivoxel pattern analysis, which uncovered multi-taste-type representations in the insula

(Fig. 4e, f), replicating our findings above and consistent with gustatory cortex representing basic taste qualities.

## Discussion

The present study employed multivoxel pattern analysis of taste-type coding, providing evidence of human gustatory cortex localized to the insula and overlying operculum. These taste quality representations were associated with but not entirely explained by palatability[12], i.e., the trial-by-trial hedonic responses. These findings were not only replicated in a separate cultural and ancestral population sample but using new compounds that elicit similar taste qualities, suggesting that these gustatory cortical representations discriminate the sensory experience of taste types rather than specific chemicals or receptor types.

In comparison with rodent studies[3] that show distinct activation clusters of basic tastes, our results revealed a more complex gustotopic map represented in the human brain of multi-taste-type representations. This complexity is reflected in the ongoing debate between the two major theories of taste coding. The "labeled line" model posits that single-taste-type information is coded by a dedicated set of receptor cells specifically tuned for that taste and relayed to the central nervous system via taste-specific afferent fibers. In contrast, the "across fiber pattern" theory posits that information is transmitted across multiple afferent fibers coding taste-type information via population spatio-temporal pattern codes. Although, at the level of taste receptor, recent evidence favors "labeled line" theory[32–35], whether such clear distinction exists in the gustatory cortex is still controversial, especially in humans[14]. Our examinations do not resolve either position in the taste-coding debate and neither was our hypothesis targeted to uncover the presence of one-taste-one-neuron, given the limitations of fMRI. Hemodynamic responses measured by fMRI may reflect relatively distant neuronal responses[36], unlike two-photon calcium imaging in rodents[3]. Furthermore, even if single-taste-coding neurons were present, we could not differentiate them if they were proximal to other taste-coding neurons, given that the fMRI blood-oxygen-level dependent (BOLD) response is orders of magnitude coarser in resolution than single neurons.

We also observed inter-individual differences in classification performance (Fig. 2). Rather than random variation, this may be explained by individual differences in taste discrimination, as suggested by a recent study that showed poorer taste classification performance of cortical signals (electroencephalography) for participants who showed poorer behavioral ability to distinguish between tastes[37]. Although we could not test this hypothesis for lack of taste discrimination testing, it poses an important future avenue for refining the individual differences in gustatory cortical representations of taste types.

Although the human insula is commonly recruited across many domains, it notably contains primary interoceptive cortex, which receives afferents from visceral organs such as the gut[38]. These separate projections from distinct sensory receptors, such as gastric distension, temperature, or pain[38], may overlap with projections from chemosensory receptors of the tongue and oral cavity to constitute a greater interoceptive whole. This positioning may suggest gustation as a homeostatic interface between the inside and outside of the body, providing a compass of what and what not to incorporate for sustenance and the changing needs of the body.

Our identification of multi-taste representations may overlap with more anterior insular regions associated with the re-representation of interoceptive and gustatory inputs, combining information from the internal and external senses[38]. These multi-taste representations may afford comparison of different tastes on

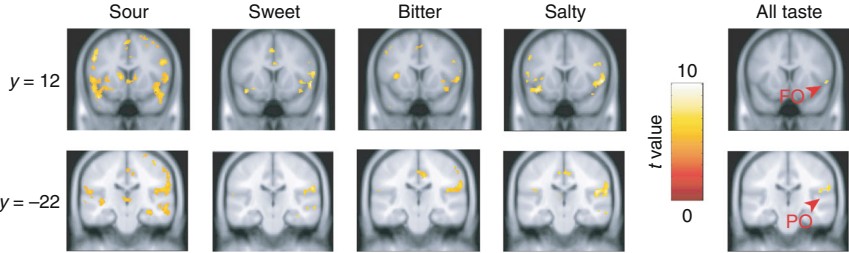

**Fig. 3** Multivoxel patterns supporting taste types outside the insula. Whole brain searchlight analysis revealed the frontal and parietal operculum also discriminate four taste types ($n = 20$ participants). FO: frontal operculum, PO: parietal operculum

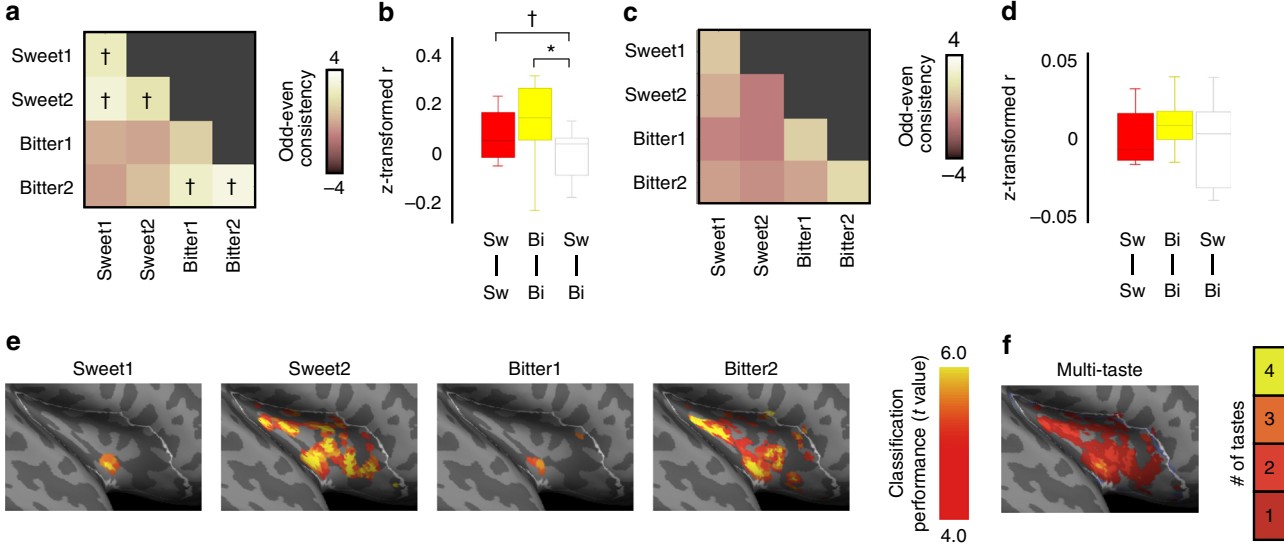

**Fig. 4** Specificity of taste quality not chemical compound in the insula under super-high field strength. **a** Correlations of voxel activation between odd and even runs between all taste combinations within each participant, submitted to one-sample $t$-test across participants ($n = 11$ participants). Statistically significant correlations within taste types but not across taste types suggest voxel-specific tuning at the individual level. **b** Correlations between odd and even runs for each same taste and different taste combinations in **a** show within-taste-type correspondence. Boxes represent the median and 25th/75th percentiles and whiskers represent the minimum and maximum. **c** The same correlations in **a** computed for voxels defined by group-level sensitivity to taste stimuli for each subject in a leave-one-out procedure. One-sample $t$-test across participants showed no taste specificity at the group level. **d** Correlations between odd and even runs for each same taste and different taste combinations in **c** reiterate lack of group-level taste specificity. Boxes represent the median and 25th/75th percentiles and whiskers represent the minimum and maximum. **e** Multivoxel pattern analysis shows group-level discriminability maps for sweet 1, sweet 2, bitter 1, and bitter 2 stimuli. **f** The number of taste stimuli represented in the insula. This replicates multi-taste population codes representing and discriminating multiple tastes in the anterior/middle insula as is shown in Fig. 2b. †$p < 0.05$ uncorrected, *$p < 0.05$ after Bonferroni correction for multiple comparisons (FWE < 5%)

a common hedonic scale, aligned with more abstract emotional experiences, such as disgust[2, 38] and delight[1]. The convergence of individual gustatory maps of basic taste qualities into a multi-taste code, in conjunction with other modalities such as smell and texture, may support the complex experiences of palatability and flavor.

## Methods

**Subjects and imaging procedures.** Experiment 1 (3.0 T): Twenty healthy adults (11 male, ages 26.2 ± 3.1 years) provided informed consent to participate in the experiment. The data of 16 participants have already been reported in our previous study[15]. Four participants were newly recruited for this study. Exclusion criteria include significant psychiatric or neurological history. This study was approved by University of Toronto Research Ethics Board (REB) and SickKids hospital REB. No statistical test was run to determine sample size a priori. The sample sizes we chose are similar to those used in previous publications[16, 26]. The experiment was conducted using a 3.0 T fMRI system (Siemens Trio). Localizer images were first collected to align the field of view (FOV) centered on each participant's brain. T1-weighted anatomical images were obtained (1 mm³, 256 × 256 FOV; MPRAGE sequence) before the experimental echo-planar imaging (EPI) runs. For functional imaging, a gradient echo-planar sequence was used (repetition time

(TR) = 2000 ms; echo time (TE) = 27 ms; flip angle = 70°). Each functional run consisted of 263 whole-brain acquisitions (40 × 3.5 mm slices; interleaved acquisition; FOV = 192 mm; matrix size = 64 × 64; in-plane resolution of 3 mm). The first four functional images in each run were excluded from analysis to allow for the equilibration of longitudinal magnetization.

Experiment 2 (7.0 T): Eleven healthy adults (6 male, ages 22.2 ± 2.2 years) provided informed consent to participate in the experiment. This study was approved by ethical committee of the National Institute for Physiological Sciences of Japan. No statistical test was run to determine sample size a priori. The sample sizes we chose are similar to those used in previous publications[16, 26]. The experiment was conducted using a 7.0 T fMRI system (Siemens Magnetom). Localizer images were first collected to align the FOV centered on each participant's brain. T1-weighted anatomical images were obtained (0.75 mm isometric, 224 × 300 FOV; MPRAGE sequence). For functional imaging, a gradient echo-planar sequence was used (TR = 500 ms; TE = 25 ms; flip angle = 35°; multiband factor = 4). Each functional run consisted of 1010 whole-brain acquisitions (32 × 2.0 mm slices; interleaved acquisition; FOV = 208 mm; matrix size = 104 × 104; in-plane resolution of 2 mm). The first four functional images in each run were excluded from analysis to allow for the equilibration of longitudinal magnetization.

**Behavioral procedures.** Experiment 1: Gustatory stimuli were delivered by plastic tubes converging at a plastic manifold, whose nozzle dripped taste solutions into

the mouth. Tubes were prefilled such that almost no delay between the visual stimulus presentation and the liquid delivery was observed. One hundred taste solution trials were randomized and balanced across five runs. In each trial, 0.5 mL of taste solution was delivered over 1244 ms. When liquid delivery ended, a screen instructed participants to swallow the liquid (1 s). After 7756 ms, scaling bars appeared to rate positivity (3 s) then negativity (3 s) of the liquid. This was followed by 0.5 mL of the tasteless liquid delivery during 1244 ms for rinsing, followed by the 1 s swallow instruction. After a 7756 ms inter-trial-interval, the next trial began.

Experiment 2: In comparison with Experiment 1, gustatory stimulus delivery differed only in their timing and quantity. One hundred taste solution trials were randomized and balanced across five runs. In each trial, 0.88 mL of taste solution was delivered over 2 s. When liquid delivery ended, a screen instructed participants to swallow the liquid (2 s). After 4000 ms, scaling bars appeared to rate positivity (3 s) then negativity (3 s) of the liquid. This was followed by 0.88 mL of the tasteless liquid delivery during 2 s for rinsing, followed by the 2 s swallow instruction. After a 7 s inter-trial-interval, the next trial began.

**Pre-experimental session**. Experiment 1: To account for individual differences in their subjective experiences of different tastes, participants were asked to taste a wider range of intensities (as measured by molar concentrations) of the different taste solutions (sour, salty, bitter, and sweet). In this pre-experimental session, participants were tested for 1 trial (2 mL) of each of the 16 taste solutions as follows: (1) sour/citric acid: $1 \times 10^{-1}$ M, $3.2 \times 10^{-2}$ M, $1.8 \times 10^{-2}$ M, and $1.0 \times 10^{-2}$ M; (2) salty/table salt: $5.6 \times 10^{-1}$ M, $2.5 \times 10^{-1}$ M, $1.8 \times 10^{-1}$ M, and $1.0 \times 10^{-1}$ M; (3) bitter/quinine sulfate: $1.0 \times 10^{-3}$ M, $1.8 \times 10^{-4}$ M, $3.2 \times 10^{-5}$ M, and $7.8 \times 10^{-6}$ M; and (4) sweet/sucrose: 1.0 M, 0.56 M, 0.32 M, and 0.18 M. The order of presentation was randomized by taste and then by concentration within each taste. After drinking each solution, participants rinsed and swallowed 5 mL of water, then rated the intensity and pleasantness (valence) of the solution's experience on separate scales of 1–9. The concentrations for each taste that matched in intensity were selected. Previous work[2] had shown that participants have different rating baselines and the concentrations most reliably selected are above medium self-reported intensity. All solutions were mixed using pharmaceutical grade chemical compounds from Sigma Aldrich (http://www.sigmaaldrich.com), safe for consumption.

Experiment 2: Participants were tested for 1 trial (1 mL) of each of the 16 taste solutions as follows: (1) sweet 1/glucose: 2.0 M, 1.1 M, 0.56 M, and 0.38 M; (2) sweet 2/sucralose: $2.1 \times 10^{-3}$ M, $1.1 \times 10^{-3}$ M, $0.53 \times 10^{-4}$ M, and $0.26 \times 10^{-4}$ M; (3) bitter 1/catechin: $3.5 \times 10^{-2}$ M, $1.8 \times 10^{-2}$ M, $8.8 \times 10^{-3}$ M, and $4.4 \times 10^{-3}$ M; and (4) bitter 2/magnesium chloride: 0.4 M, 0.2 M, 0.1 M, and 0.05 M. The order of presentation was randomized by taste and then by concentration within each taste. After drinking each solution, participants rinsed and swallowed 5 mL of water, then rated the intensity and pleasantness (valence) of the solution's experience on separate scales of 1–9. The concentrations for each taste that matched in intensity were selected. All solutions were mixed using food-grade chemical compounds from DHC (catechin), FUJIFILM Wako Pure Chemical Corporation (magnesium chloride), Tsuruya Chemical Industries (sucralose), and Nichiga (glucose).

**Data analysis**. Data were analyzed using SPM8 software (http://www.fil.ion.ucl.ac.uk/spm/). Functional images were realigned, slice timing corrected, and normalized to the MNI template (ICBM 152) with interpolation to a $2 \times 2 \times 2$ mm space. Data were spatially smoothed (full width, half maximum (FWHM) = 6 mm) for univariate parametric modulation analysis but not for multivoxel pattern analysis as it may impair performance[19]. Each stimulus presentation was modeled as a separate event, using the canonical function in SPM8. For each voxel, $t$-values of individual trials were demeaned by subtracting the mean value across trials. To visualize the imaging results, freesurfer software[39] (http://surfer.nmr.mgh.harvard.edu/) and SPM surfrend toolbox (written by I. Kahn; http://spmsurfrend.sourceforge.net) were used after modification.

**Univariate analysis**. We conducted univariate analyses to examine whether basic tastes were coded by specific voxels in the insula. Regressors coding each tastant were time-locked to stimulus presentation. Univariate analyses were conducted twice: with and without hedonic valence regressors (Fig. 1). To visualize how much variance could be explained by hedonic valence regressors in the bitter sensitive regions, we selected significant voxels in the contrast "bitter vs. tasteless" without valence regressed out (Supplementary Figure 1). Averaged activity was shown for activity against resting baseline (Supplementary Figure 1a), activity against tasteless (Supplementary Figure 1b), and activity against tasteless with valence regressed out (Supplementary Figure 1c). To test the existence of voxel-specific taste tuning, we split each participant's odd and even runs, comparing the voxel activity for each taste in the odd runs to the voxel activity for each taste in the even runs. For illustration, when voxels were rank-ordered based on activation to each taste in even runs, we found consistent patterns of correspondingly decreasing activation for all four tastes in odd runs (Fig. 1b, Supplementary Figure 4). Correlations were computed for voxel activation between odd and even runs between all taste combinations within each participant, submitted to one-sample $t$-test across participants (Fig. 1c, Fig. 4a, b). We further computed correlations between odd and even runs for all same taste and different taste combinations within each

participant. Correlation coefficients were $z$-transformed and subject to one-sample $t$-test across participants (Fig. 1d, Fig. 4b, d).

**Searchlight analysis for taste-type representations**. We analyzed fMRI data of the insular cortex using searchlight (radius of 4 mm, including 33 voxels) analysis[20]. Within a given sphere for each participant, a vector was created containing the spatial pattern of BOLD-MRI signal time-locked to stimulus presentation (normalized $t$-values per voxel). To evaluate whether the activity patterns in the searchlight spheres are capable of discriminating taste types, we employed a leave-one-stimulus-pair-out cross-validation[40]. In this procedure, the LDA classifier was trained on 38 trials, which included the tested taste type and another taste type (19 trials for each) and then tested on the left-out stimulus pair. Classification performance for each taste was averaged across comparisons with other tastes (e.g., sour classification performance was averaged across sour vs. sweet, sour vs. bitter, sour vs. salty, and sour vs. tasteless). At the level of individuals, 58.7% classification accuracy corresponded to $p < 0.05$ uncorrected. For group analysis, the individual classification performance maps were smoothed with a 4 mm FWHM Gaussian kernel and then subjected to a one-sample permutation test using SnPM13 (http://warwick.ac.uk/snpm). In this procedure, the data from each participant were randomly flipped by multiplying $-1$ after subtracting 50% (chance level accuracy) and then subjected to a one-sample $t$-test across participants. This was permuted 10,000 times, resulting in the distribution of maximal $t$ within the insula. Based on this distribution, the 5% FWE threshold was determined.

**Taste conjunction analysis**. For multi-taste conjunction analysis (Fig. 2b), each voxel was tested on whether it exceeded the threshold for the four taste-type discriminations where each taste-type discrimination averaged the classification performance across four comparisons (e.g., sour vs. sweet, sour vs. bitter, sour vs. salty, and sour vs. tasteless), exceeding chance classification (50%). Valid conjunction inference requires significant results for all comparisons[41]. We thus counted the number of taste types satisfying 5% FWE threshold in each voxel within the insula.

**Taste-pair analysis**. For analysis of specific taste pairs, we examined an independently defined ROI within the insula. First, we used a leave-one-subject-out procedure, excluding each of 20 subjects, then computing a 4-taste conjunction map (i.e., voxels satisfying all 4 taste contrasts described above) with the remaining 19 subjects, resulting in 20 maps. The overlap of these 20 group maps is shown in Fig. 2c. The voxels from the map satisfying 5% FWE threshold was defined as the ROI capable of taste discrimination. Within this ROI, we examined discrimination of specific taste pairs, computing classification performance of each taste pair. Group performance was computed as the average classification performance across 20 subjects (Fig. 2d).

For taste-pair discrimination based on ratings of valence (Fig. 2e), we conducted an LDA analysis using subject ratings of valence (i.e., independent of fMRI data). Valence was calculated by subtracting negativity rating from positivity rating for each trial. Taste classification was computed using a leave-one-trial-out trained on the 19 remaining trials for each taste type.

**Valence and taste-type analysis**. To test the independence of taste type from valence, we examined the similarity of fMRI data within the ROI defined by the four-taste conjunction map above. For each subject, trial-by-trial correlations were calculated, resulting in 4950 ($100 \times 99/2$) correlation coefficients, sorted into $2 \times 2$ categories of taste type (same, different) $\times$ hedonic valence (same, different). The correlation coefficients were averaged within each cell per subject, then all subjects' data were subjected to a two-way repeated-measures ANOVA with taste type and valence as factors (Fig. 2g).

We further conducted a follow-up analysis with no data dependency in the trial-by-trial correlations across the $2 \times 2$ cells. We randomly populated each cell with the 100 trials, repeating this procedure 1,000,000 times. From these, we analyzed the permutation with the greatest congruence between the $2 \times 2$ factors and actual trial categories (based on the maximum geometric mean of the proportion of the reduced data across the four cells). We then computed cross-trial correlations only within each cell, ensuring no cross-cell dependency. The correlation coefficients were averaged within each cell per subject, then all remaining subjects' data were submitted to a two-way repeated-measures ANOVA (Supplementary Figure 3).

Taste-type patterns differed independently of taste valence, i.e., taste discrimination maps were dissociable from palatability. Due to the strong association between taste type and valence, trial combinations were not equal across the levels. For instance, same valence with different taste types were relatively rare (Supplementary Table 4). However, this does not indicate multicollinearity in effect size.

**Statistics**. We analyzed the data without assuming normal distribution, using non-parametric statistics. Before ANOVA (Fig. 2g), Levene's test was conducted to ensure the assumption of homoscedasticity was met. Multiple comparison corrections were applied, using Bonferroni correction.

**Reporting summary**. Further information on experimental design is available in the Nature Research Reporting Summary linked to this article.

## Data availability
All relevant data and code used to generate results are available from the authors on request.

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

## Acknowledgements
We thank R. Markello, M. Taylor, M. Misaki, D. Hamilton, and K. Gardhouse for technical collaboration and discussion. We also thank M. Fukunaga for supervision for super-high field MRI data collection. This work was funded by CIHR to A.K.A. and JSPS KAKENHI Grant Number JP17H06033 and the grant from the Takeda Science Foundation to J.C.

## Author contributions
J.C. and A.K.A. designed the experiments. J.C. and D.H.L. built the experimental apparatus and performed the experiments. J.C. analyzed the data. J.C., D.H.L., N.K. and A.K.A. wrote the paper. N.K. and A.K.A. supervised the study.

## Additional information

**Competing interests:** The authors declare no competing interests.

**Journal Peer Review Information**: *Nature Communications* thanks Dana Small and other anonymous reviewer(s) for their contribution to the peer review of this work. Peer reviewer reports are available.

