## [Peer Review File · Nature Communications]

Reviewers' comments:

Reviewer #1 (Remarks to the Author):

This is an interesting manuscript examining the representation of taste in the mammalian cortex. Unfortunately, the resolution is very low, and as such it is difficult to assess the conclusions: are there separate representation, but intermingled? Are there separate representations and segregated, but the resolution does not allow fine separation?.

I have one major problem and a couple of suggestions:

Major:

- The use of a single bitter (quinine) for the study is a serious problem. This is particularly relevant as quinine is known to affect the function of neurons, and there are plenty of neuronal terminals in the oral cavity. Therefore, the "bitter" activity they observed can be the result of multiple inputs. Fortunately, there are many dozen receptors for bitter in humans, so the best way to eliminate potential issues associated with quinine would be to use a second bitter.

Suggestions:

- The rodent imaging data suggests strong topography for sweet and bitter at the opposite boundaries of insula cortex (A-P axis). The authors should perform some of their analysis on these separate domains, where perhaps the limited spatial resolution of fMRI could still provide some clarity. Again, using at least two different bitter tastants.

- The fundamentals behind the analysis to remove confounds from hedonic value associated with the different tastes needs significant rewrite and clarity (in the text).

Reviewer #2 (Remarks to the Author):

This is a potentially very important paper addressing the location of the human gustatory cortex and the existence of chemotopy. The authors use multivariate pattern analysis combined with a searchlight approach to sampling insular cortex to reveal an area of mid central insula that shows consistent pattern discrimination of sweet, sour, salty and bitter tastants, with no evidence for chemotopy at this spatial resolution. They also attempt to disentangle hedonic from quality coding using post-processing techniques.

The MVPA approach is elegant and the primary findings localizing a core gustatory region is beyond reproach. I am less convinced by the argument that they have definitively dissociated quality from valence, but I also do not believe that this is central to the value of the paper as past work in humans and animals (with one exception) also fail to adequately address this issue. Here my primary concern is that there is a multicollinearity issue with quality and valence. In addition, the 2 * 2 repeated measures analysis where patterns are sorted into same, different for quality and valence is, in my opinion, falls short because there is inherently more variability in same/different for valence which will include multiple stimuli (e.g. bitter/salt) compared to same/different for quality, which will capture the same stimulus. The authors have also not considered physiological significance as a potentially additional factor beyond valence. I think it would be really helpful to include the raw valence ratings and to evaluate how consistent these ratings are WITHIN subjects.

While I appreciate the brevity of the journal style, the authors should absolutely discuss the labeled

line vs. across fiber pattern theory controversy as the work is central to this issue.

Another important debate in the field that is not adequately addressed is the controversy over the importance of the parietal and frontal operculum in taste coding, with some (e.g. Kobayakawa in the late 90s and Ogawa et al., 2004) arguing that the primary taste region resides here whereas others argue that it is in the mid-insula (consistent with the current findings) based on the observation that the region is sensitivity to baseline shifts of attention (Veldhuizen et al., 2007). Relatedly, it would very informative if the authors could expand their searchlight to include operculum so that they can weigh in more definitively on this issue.

How do the authors rule out differences in swallowing the stimuli (e.g. bitter vs. sweet) as contributing to effects?

What role might hedonic contrast play in driving valence ratings?

Minor

Why 4mm radius for the searchlight?

Line 39-40 – what is taste appetitive or aversive quality? This needs further explanation!

Line 123. Please explain further the procedure “using other subjects’ data as applied to this specific analysis.

Line 127-129. This section is unclear.

Line 219. reference 51 does not exist (copy and paste error?).

The figures are really nice!

Reviewer #3 (Remarks to the Author):

With this manuscript, Chikazoe et al. aim to delineate the spatial mapping of taste qualities in the putative primary gustatory area of the insular cortex.

This question is of high relevance given that a topographical organization has been described for most sensory modalities, but has not yet been conclusively determined for taste or smell.

A particular strength of the paper is that authors use a variety of complimentary neuroimaging methods such as GLM, MVPA and RSA to define spatial distributions. It is also noteworthy that they address an important potential confound, the association between taste quality and valence.

As such, I think the paper constitutes a novel contribution to a wider field of perceptual neuroscience, which will appeal to the readers of nature communications. I think that the paper would, however, benefit from a number of clarifications, which I outline below. In particular, I think that the paper would benefit from a more thorough placement into the context of the goal of identifying a topographic map and explicit evaluation of where the findings leave us with respect to that goal.

- the procedures for the univariate analysis should be clarified with respect to their relationship to the voxel-specific tuning of specific taste modalities. To my understanding, the contrasts depicted in the

first three rows of figure 1 depict activation for each taste modality against a low-level baseline. As such, if I understand correctly, overlapping activations between the four modalities are possible and the activations present no evidence for specificity of voxel tuning to one of the taste modalities over the others. I would expect to see the taste qualities contrasted against each other for evidence of voxel-specific tuning.

- In addition, it appears that the authors find no taste activation relative to a tasteless stimulus baseline for any of their stimuli, which is somewhat surprising given that the majority of taste neuroimaging studies do report significant univariate activation in these areas, as summarized in the cited metaanalysis by Veldhuizen et al. It would be helpful if the authors reported wholebrain activation of taste stimulation relative to a no-taste baseline or made this information available as supplementary material to allow the reader to get an idea of the overall strength of gustatory signal acquired through their stimulation method.

- I am missing a description of the gustometer that delivered the stimuli, how it was controlled and time-locked to the visual stimuli as well as to the TR. Presumably, if triggering of the tastant occurred outside the scanner room, a certain delay had to be taken into account?

- The authors should consider putting the data from the two subjects in figure 2a on the same scale for comparability-if not possible due to large inter-subject variability in classification performance, this might be something interesting to comment on. A recent study that used multivariate analysis of EEG signal to investigate tastant-specific cortical coding (Crouzet, Busch & Ohla, (2015), Current Biology) reported poorer classification performance of the cortical signal for participants who showed poorer behavioral ability to distinguish between tastants. If the authors have individual discrimination performance measures available it would be interesting to see if there is a link between the two here as well, as evidence for a direct functional relevance for the separability of the neural signal.

- Given that the authors' main aim is to establish a topographic map of gustatory stimulation, I think that it would be appropriate to speculate a bit more extensively on what such a topographic organization would look like. They state, "Our finding of increasing numbers of taste type coded along the posterior to anterior axis of the insular cortex is most consistent with the mid-posterior insula containing regions of neurons tuned to primary sensory taste qualities". I find the assumption of an increase on a posterior to anterior axis a bit difficult to follow based on the figures, which indicate to me a restricted core area that is included in representation of all four tastes, and a abrupt drop to one or two taste discrimination outside this area in every topographical direction, not just posteriorly. On the same note, what is meant by "discrete" and "multi-taste-type" representations could also be clarified more explicitly- is the assumption that any voxel that is involved in classification of all tastants is therefore unspecific and irrelevant for topographical representation of taste quality? The authors state in the abstract of the paper that they identified "insular regions specifically sensitive to Sweet, salty, bitter, and sour tastes". This creates an expectation of a topographical map for the different tastants, which the paper then fails to provide. This should be made clear from the abstract, and the implications of this and differences from other sensory modalities discussed more explicitly throughout the manuscript.

- Given that the answer format consisted of two linear scales for positivity and negativity of the stimulus, the authors should describe how they converted this information into a binary score for same vs different hedonic valence.

- Line 107. I don't understand what the authors mean by the sentence starting with "Objective differences..." given that there is no direct evidence that the observed signal is derived from accumulated signal across the entire tongue, and also, I understand the data to indicate that no

discrete regions for each sensory modality were found. It would help to clarify this by providing additional context to the statement.

- I would recommend a revision of the description of the analysis for assessment of cross-subject correspondence to enhance readability and clarity. Specifically, it is unclear which part of figure 2a such a map would be similar to. Are they referring to the group statistic here? I interpret figure 2c to indicate for each voxel, how many of these individual subject masks it was included in. I don't understand though what the term "significant maps" means in this context. I also find it difficult to visually extract meaningful voxel-level information out of figure 2c. For getting an idea of the number and location of overlapping voxels between subjects, an alternative presentation, possibly on a rendered surface like in the images above, would be more informative. It is quite interesting, however, to note on picture 2c that the left insular cluster appears much smaller and outside the dorsal anterior insula, which is commonly described as primary gustatory cortex. Given the topographical difference I would be curious to know if there is a classification accuracy difference between the two hemispheres as well.

- The final paragraph of the discussion is a bit surprising since it takes up a variety of topics that were not previously discussed in the manuscript such as interoception, flavor, and palatability, as well as secondary "re-representations". While these concepts in themselves are highly interesting, their relevance in the context of the present study should be more clearly explained.

Reviewer #1 (Remarks to the Author):

This is an interesting manuscript examining the representation of taste in the mammalian cortex. Unfortunately, the resolution is very low, and as such it is difficult to assess the conclusions: are there separate representation, but intermingled? Are there separate representations and segregated, but the resolution does not allow fine separation?

1) The use of a single bitter (quinine) for the study is a serious problem. This is particularly relevant as quinine is known to affect the function of neurons, and there are plenty of neuronal terminals in the oral cavity. Therefore, the "bitter" activity they observed can be the result of multiple inputs. Fortunately, there are many dozen receptors for bitter in humans, so the best way to eliminate potential issues associated with quinine would be to use a second bitter.

Following the reviewer's suggestion, we conducted another experiment (exp. 2) in which we used 2 new bitter tastants (MgCl and catechin) and 2 new sweet tastants (glucose and sucralose). Further, to clarify whether the multi-taste-type representations resulted from the low resolution of functional MRI, we now employ super-high magnetic field MRI scanner (7T) located in National Institute for Physiological Sciences in Japan. With these additions, we have replicated the main findings from exp. 1.

2) The rodent imaging data suggests strong topography for sweet and bitter at the opposite boundaries of insula cortex (A-P axis). The authors should perform some of their analysis on these separate domains, where perhaps the limited spatial resolution of fMRI could still provide some clarity. Again, using at least two different bitter tastants.

As the reviewer suggested, we examined this A-P axis separation of bitter and sweet regions in the insula in the additional experiment that provided greater voxel resolution. We did not find this A-P axis separation.

We note that the higher resolution in experiment 2 did show voxel-level consistency of taste types within participants (univariate analyses showing voxels sensitive to bitter in even runs were sensitive to bitter in odd runs). However, this was not present at the group level. Group level discrimination of taste types in experiment 2 was demonstrated using multivoxel analyses, as in experiment 1. We provided a new figure to demonstrate these results (Fig 4).

The replication of our main finding of multi-taste type representations rather than spatially discrete representations under improved spatial and temporal resolution suggests something greater than a technological discrepancy between rodent studies and the present study. However, we note that functional MRI has limitations, as it cannot directly observe neuronal activity, but rather, hemodynamic responses associated with neuronal responses. As is discussed in classical fMRI papers (e.g. Turner 2001), the fMRI activation measured by gradient echo sequence may reflect the neuronal activity in the distant region, thus, we cannot conclude that the taste representations are intermingled. We highlight this limitation of fMRI in the discussion section (line 212 - 216).

We emphasize that this is the first neuroimaging study which demonstrates the distinct taste representations in the insula, while noting that future studies will be necessary to precisely address the differences between rodents and humans under fMRI.

3) The fundamentals behind the analysis to remove confounds from hedonic value associated with the different tastes needs significant rewrite and clarity (in the text).

We appreciate the reviewer's critique. We have revised the text so that the fundamentals behind this analysis are clearer for general readers (line 150 – 155). As taste quality and palatability are often coupled, it is unclear whether taste type discrimination is explained by hedonics. In our previous study (Chikazoe et al., 2014), we demonstrated that representational similarity could be approximated by linear summation of spatial pattern correlations of multiple components (e.g. taste types and valence). Based on this assumption, we examined whether correlations between trials can be decomposed into 2 components, that is, taste types and valence.

Reviewer #2 (Remarks to the Author)

1) The MVPA approach is elegant and the primary findings localizing a core gustatory region is beyond reproach. I am less convinced by the argument that they have definitively dissociated quality from valence, but I also do not believe that this is central to the value of the paper as past work in humans and animals (with one exception) also fail to adequately address this issue. Here my primary concern is that there is a multicollinearity issue with quality and valence. In addition, the 2 * 2 repeated measures analysis where patterns are sorted into same, different for quality and valence is, in my opinion, falls short because there is inherently more variability in same/different for valence which will include multiple stimuli (e.g. bitter/salt) compared to same/different for quality, which will capture the same stimulus.

We agree multicollinearity of quality and valence is an important issue. Regarding the 2 x 2 ANOVA, we note that from the aspect of number of trial combinations, we found a strong association between taste type and valence: Most of the same valence combinations come from the same taste type combinations while the same valence combinations with the different taste types are relatively rare (see newly added Supplementary Table S3). However, this does not indicate multicollinearity in effect size. The 2 x 2 ANOVA revealed that taste type, but not valence, explains most of the variance of activation pattern similarity (Fig 2g). We further conducted an alternate 2 x 2 ANOVA that mitigates the issue of data dependency (Fig S3). These analyses are now elaborated upon in the methods section (line 416 - 436).

2) I think it would be really helpful to include the raw valence ratings and to evaluate how consistent these ratings are WITHIN subjects.

We now provide a summary table showing how stable/unstable valence ratings are within subjects. For each taste, we calculated the mode of valence ratings and counted the number of trials showing the different valence ratings from the mode (Supplementary Table S1).

3) While I appreciate the brevity of the journal style, the authors should absolutely discuss the labeled line vs. across fiber pattern theory controversy as the work is central to this issue.

We now discuss how our findings should be interpreted from the context of the labeled line vs. across fiber pattern theory (line 210 - 222).

4) Another important debate in the field that is not adequately addressed is the controversy over the importance of the parietal and frontal operculum in taste coding, with some (e.g. Kobayakawa in the late 90s and Ogawa et al., 2004) arguing that the primary taste region resides here whereas others argue that it is in the mid-insula (consistent with the current findings) based on the observation that the region is sensitivity to baseline shifts of attention (Veldhuizen et al., 2007). Relatedly, it would very informative if the authors could expand their searchlight to include operculum so that they can weigh in more definitively on this issue.

We now provide the whole brain map for the searchlight analysis showing taste type representations in the frontal and parietal operculum (see new Figure 3), as well as including discussion of the role of the human gustatory cortex in taste quality and valence coding (line 202 - 209).

5) What role might hedonic contrast play in driving valence ratings?

We analyzed the effect of hedonic contrast (i.e. bitter following sweet and vice versa) on valence ratings and found that the hedonic contrast did not affect ratings. This may be due to a long interstimulus interval (~30s). We have included in the text and the result of this analysis, suggesting it does not account for variability in valence ratings (line 89 - 92).

Minor comments

6) Why 4mm radius for the searchlight?

We assumed that taste type representations may be spatially separated in the insula, and that they might be detected only by MVPA. The use of a greater searchlight radius (e.g. 6mm) may have underestimated the spatial separation between taste sensitive regions. To avoid this, we employed relatively smaller radius (i.e. 4mm) for the searchlight, which is now stated in the text (line 127 - 128).

7) Line 39-40 – what is taste appetitive or aversive quality? This needs further explanation.

Taste appetitive or aversive quality relates to the palatability of the gustatory stimuli. We have revised the text to make this distinction clearer (line 83 -86).

8) Line 123. Please explain further the procedure “using other subjects’ data as applied to this specific analysis.

This is a version of leave-one-out procedure to ensure independence of fMRI signal characterization from ROI definition. First, we left a subject out and then created the functional ROI defined by the overlap of all the taste representations, using the other 19 subjects’ data. Using this ROI, the confusion matrix for taste discriminability was made for the left-out subject. This procedure was repeated such that each subject is used once as the test data. By averaging these confusion matrices, we obtained ROI analysis results while avoiding the circularity issue (the use of the same dataset for selection and selective analysis). We now make this clear in the text (line 136 - 149).

9) Line 127-129. This section is unclear

We have deleted this portion of the text.

10) Line 219. reference 51 does not exist (copy and paste error?).

Thank you for catching our error. We have corrected it.

Reviewer #3 (Remarks to the Author):

1) The procedures for the univariate analysis should be clarified with respect to their relationship to the voxel-specific tuning of specific taste modalities. To my understanding, the contrasts depicted in the first three rows of figure 1 depict activation for each taste modality against a low-level baseline. As such, if I understand correctly, overlapping activations between the four modalities are possible and the activations present no evidence for specificity of voxel tuning to one of the taste modalities over the others. I would expect to see the taste qualities contrasted against each other for evidence of voxel-specific tuning.

As the reviewer suggested, overlapping activations between the four taste types are possible. However, after regressing out valence, the taste vs. tasteless contrast did not survive multiple comparisons. We found the same null results for each taste contrasted against each other taste (e.g. sour vs. bitter).

To further examine the existence of voxel-specific tuning, we now report additional univariate analyses. Here we split each participant's even and odd runs, examining the similarity (correlation) in voxel activity between tastes between runs. In the first experiment (3T), we found voxels sensitive to one taste were sensitive to the other tastes (i.e., showing no voxel-specific taste tuning within participants; Fig 1b,c,d). The same analysis in the second experiment (7T) showed within-taste type but not cross-taste type voxel-specificity within subjects (Fig 4a,b, Fig S4a), which was encouraging. However, this disappeared at the group level (Fig 4c,d, Fig S4b). These results further suggest high inter-individual topographical variability but support for region specific taste tuning with an individual.

2) In addition, it appears that the authors find no taste activation relative to a tasteless stimulus baseline for any of their stimuli, which is somewhat surprising given that the majority of taste neuroimaging studies do report significant univariate activation in these areas, as summarized in the cited metaanalysis by Veldhuizen et al. It would be helpful if the authors reported wholebrain activation of taste stimulation relative to a no-taste baseline or made this information available as supplementary material to allow the reader to get an idea of the overall strength of gustatory signal acquired through their stimulation method.

We did in fact find activation for taste vs. tasteless contrasts. We had previously highlighted taste vs. tasteless controlling for valence (which revealed no activation). As the reviewer suggested, we now include whole brain activation maps (specifically contrasting bitter to tasteless without valence regression) in Supplementary Figure S2. This demonstrates the bitter-related insular activation near the peaks reported by the meta-analysis of Veldhuizen et al., suggesting the sufficient strength of the gustatory signal through our stimulation method. We also found multiple brain activation including the frontal operculum, postcentral gyrus/parietal operculum, thalamus and cingulate gyrus, consistent with Veldhuizen et al (Supplementary Figure S2 and Supplementary Table S2). These new results have been made clear in the text (line 107 – 111).

3) I am missing a description of the gustometer that delivered the stimuli, how it was controlled and time-locked to the visual stimuli as well as to the TR. Presumably, if triggering of the tastant occurred outside the scanner room, a certain delay had to be taken into account?

In our setting, almost no delay between the visual stimulus presentation and the liquid delivery was observed. This is possible because each tubing is pre-filled with its tastant up to the manifold on the participant's tongue, such that a trial's trigger delivers the liquid to the participant immediately. We have clarified this in the text (line 286 - 288).

4) The authors should consider putting the data from the two subjects in figure 2a on the same scale for comparability-if not possible due to large inter-subject variability in classification performance, this might be something interesting to comment on. A recent study that used multivariate analysis of EEG signal to investigate tastant-specific cortical coding (Crouzet, Busch & Ohla,(2015), Current Biology) reported poorer classification performance of the cortical signal for participants who showed poorer behavioral ability to distinguish between tastants. If the authors have individual discrimination performance measures available it would be interesting to see if there is a link between the two here as well, as evidence for a direct functional relevance for the separability of the neural signal.

Following the reviewer's suggestion, we now use the same scale for the 2 subjects in Figure 2a. Unfortunately, because we did not have data for individual discrimination performance, we could not analyze the relationship between behavioral discrimination performance and MVPA classification performance. However, we agree this is a highly interesting avenue towards a refined understanding for individual differences in cortical taste representation, and we now highlight this possibility in our discussion (line 223 - 230).

5) Given that the authors' main aim is to establish a topographic map of gustatory stimulation, I think that it would be appropriate to speculate a bit more extensively on what such a topographic organization would look like. They state, "Our finding of increasing numbers of taste type coded along the posterior to anterior axis of the insular cortex is most consistent with the mid-posterior insula containing regions of neurons tuned to primary sensory taste qualities". I find the assumption of an increase on a posterior to anterior axis a bit difficult to follow based on the figures, which indicate to me a restricted core area that is included in representation of all four tastes, and a abrupt drop to one or two taste discrimination outside this area in every topographical direction, not just posteriorly. On the same note, what is meant by "discrete" and "multi-taste-type" representations could also be clarified more explicitly- is the assumption that any voxel that is involved in classification of all tastants is therefore unspecific and irrelevant for topographical representation of taste quality? The authors state in the abstract of the paper that they identified "insular regions specifically sensitive to Sweet, salty, bitter, and sour tastes". This creates an expectation of a topographical map for the different tastants, which the paper then fails to provide. This should be made clear from the abstract, and the implications of this and differences from other sensory modalities discussed more explicitly throughout the manuscript.

For closer adherence to our findings (including the additional experiment 2), we have now removed our statements about the posterior to anterior axis. We have also removed mentions that might suggest we found a topographical map for different taste types. As the reviewer pointed out, we did

not find spatially segregated taste type representations, rather, taste type representations were overlapped. However, considering the activation patterns, taste types can be clearly discriminated in the insula, suggesting that their representations were qualitatively distinct. We now discuss this issue in the text (line 210 - 222).

6) Given that the answer format consisted of two linear scales for positivity and negativity of the stimulus, the authors should describe how they converted this information into a binary score for same vs different hedonic valence.

Valence was calculated by subtracting negativity rating from positivity rating. We now note this in the methods (line 410 - 411).

7) Line 107. I don't understand what the authors mean by the sentence starting with "Objective differences..." given that there is no direct evidence that the observed signal is derived from accumulated signal across the entire tongue, and also, I understand the data to indicate that no discrete regions for each sensory modality were found. It would help to clarify this by providing additional context to the statement.

We now clarify this in the text (line 46 - 59). In the context of our study, "objective" refers to the taste type and "subjective" refers to rated hedonic impact. We use the term objective to refer to the external sensory object rather than its resulting internal subjective response. Our analyses were confined to assess taste type stimulation rather than internal subjective response to the tastes, e.g., how bitter or sweet the subjective response. By contrast, we do quantify the subjective hedonic impact to each taste through ratings of pleasant and unpleasant valence.

8) I would recommend a revision of the description of the analysis for assessment of cross-subject correspondence to enhance readability and clarity. Specifically, it is unclear which part of figure 2a such a map would be similar to. Are they referring to the group statistic here? I interpret figure 2c to indicate for each voxel, how many of these individual subject masks it was included in. I don't understand though what the term "significant maps" means in this context. I also find it difficult to visually extract meaningful voxel-level information out of figure 2c. For getting an idea of the number and location of overlapping voxels between subjects, an alternative presentation, possibly on a rendered surface like in the images above, would be more informative.

The main group level analysis for experiment 1 is illustrated in the "Group" panel in Fig 2b, representing the conjoined 4-taste discriminability maps across subjects. This is what most closely resembles Fig 2c.

Our motivation for including Fig 2c was to show the regions from which regions the confusion matrix for pair-wise taste discrimination was computed (Fig 2d). Fig 2c represents a more stringent test of insular voxels in that the ROI for each subject left-out was determined by the remaining 19 subjects. We now clarify this in the main text (line 136 - 149).

9) It is quite interesting, however, to note on picture 2c that the left insular cluster appears much smaller and outside the dorsal anterior insula, which is commonly described as

primary gustatory cortex. Given the topographical difference I would be curious to know if there is a classification accuracy difference between the two hemispheres as well.

We have now analyzed the difference in the classification performance between both hemispheres, and found no significant differences. We now note this in the text (line 144 - 146).

Reviewers' comments:

Reviewer #1 (Remarks to the Author):

This revised manuscript addresses many of my concerns, and refocuses the paper on its main conclusion, namely demonstrating the representation of taste in insula.

A remaining (recurring) problem with the paper is the author's confusion, and apparent lack of understanding, of the meaning/difference between single-neuron coding and topography: a neuron may exhibit single taste-coding even if intermingled with other neurons also singly-tuned to other tastes. This work does not shed light into the issue of one neuron-one taste versus across-fiber coding as the resolution of these studies cannot distinguish or resolve single neuron activity (In fact, the BOLD signals may result from "activity" of neurons dedicated to a single taste, intermingled with other neurons also dedicated to single tastes, or from neurons tuned across tastes, or even mixes of both). Furthermore, since fMRI measures hemodynamic responses whose origins may be distal to the signal, even topography would be hard to assign. None of this impacts the main conclusion of the paper assigning insula, but brings a lack of clarity and rigor to the concluding section of the manuscript, even to the point that the author's propose a "massively interconnected system", so as to accommodate all possible scenarios. I strongly suggest they revise that section of the manuscript, in a thoughtful and clear way.

Reviewer #2 (Remarks to the Author):

The authors have been very responsive to reviewer comments. I am satisfied by the responses to all but one of my concerns. I believe the authors misunderstood my comment on discussing labeled line vs across fiber pattern. I was referring to these two very well known theories of how taste is coded - in a labeled line from the taste bud or across fibers as a spatial temporal code. See work by Don Katz. I do feel that this should be briefly mentioned since it is the focus of so much work in gustation and the current findings shed light on this issue.

I also think that they should note that taste-tasteless produces a nice map of taste cortex when palatability is not controlled - i.e. akin to most analyses in the literature. When one looks at figure 1 and sees no response in taste-tasteless it is misleading because it seems as though the study was underpowered or something was wrong. I think many readers will come to this conclusion as the text no stands and therefore loose confidence in the other findings.

Reviewer #3 (Remarks to the Author):

I found the resubmission to be a very thorough and thoughtful revision. All my concerns have been addressed and I have no further comments.

Reviewer #1 (Remarks to the Author):

1) A remaining (recurring) problem with the paper is the author's confusion, and apparent lack of understanding, of the meaning/difference between single-neuron coding and topography: a neuron may exhibit single taste-coding even if intermingled with other neurons also singly-tuned to other tastes. This work does not shed light into the issue of one neuron-one taste versus across-fiber coding as the resolution of these studies cannot distinguish or resolve single neuron activity (In fact, the BOLD signals may result from "activity" of neurons dedicated to a single taste, intermingled with other neurons also dedicated to single tastes, or from neurons tuned across tastes, or even mixes of both). Furthermore, since fMRI measures hemodynamic responses whose origins may be distal to the signal, even topography would be hard to assign. None of this impacts the main conclusion of the paper assigning insula, but brings a lack of clarity and rigor to the concluding section of the manuscript, even to the point that the author's propose a "massively interconnected system", so as to accommodate all possible scenarios. I strongly suggest they revise that section of the manuscript, in a thoughtful and clear way.

The ongoing debate between labeled line and across fiber pattern taste coding cannot be resolved by fMRI and as such was not a focus of our manuscript. It is difficult, if not impossible, to resolve using our methods given the nature of fMRI as noted by the reviewer. While it was not our hypotheses to discover single-taste coding neurons, we present results that provide new insights into the nature of taste representations in human insular cortex. We now further clarify these points in the discussion, regarding the limitations of fMRI and clearly contextualizing our results with respect to these two historical models of taste coding.

Pg 13-14 (lines 210-227)

“ In comparison with rodent studies³ that show distinct activation clusters of basic tastes, our results revealed a more complex gustotopic map represented in the human brain of multi-taste type representations. This complexity is reflected in the ongoing debate between the two major theories of taste coding. The “labeled line” model posits that single taste type information is coded by a dedicated set of receptor cells specifically tuned for that taste, and relayed to the central nervous system via taste-specific afferent fibers. In contrast, the “across fiber pattern” theory posits that information is transmitted across multiple afferent fibers coding taste-type information via population spatio-temporal pattern codes. Although, at the level of taste receptor, recent evidence favors “labeled line” theory³²⁻³⁵, whether such clear distinction exists in the gustatory cortex is still controversial, especially in humans¹⁴. Our examinations do not resolve either position in the taste coding debate, and neither was our hypothesis

targeted to uncover the presence of one-taste-one-neuron, given the limitations of fMRI. Hemodynamic responses measured by fMRI may reflect relatively distant neuronal responses³⁶, unlike two-photon calcium imaging in rodents³. Furthermore, even if single-taste coding neurons were present, we could not differentiate them if they were proximal to other taste-coding neurons, given that the fMRI BOLD response is orders of magnitude in resolution than single neurons.”

Reviewer #2 (Remarks to the Author):

1) The authors have been very responsive to reviewer comments. I am satisfied by the responses to all but one of my concerns. I believe the authors misunderstood my comment on discussing labeled line vs across fiber pattern. I was referring to these two very well known theories of how taste is coded - in a labeled line from the taste bud or across fibers as a spatial temporal code. See work by Don Katz. I do feel that this should be briefly mentioned since it is the focus of so much work in gustation and the current findings shed light on this issue.

We agree with the reviewer that our discussion is better served by a clearer elaboration of these two theories. This is also related to the Reviewer 1’s comment above, discussing these two theories and contextualizing our results to them. Indeed, this is a difficult question that, given our choice to use fMRI, our hypotheses were not positioned to answer. Our contributions are at a higher level of taste representation and the human gustatory cortex. Nonetheless, the theories and the position of our results to them are now stated clearly in a paragraph in the discussion on pg 13-14 (see above).

2) I also think that they should note that taste-tasteless produces a nice map of taste cortex when palatability is not controlled - i.e. akin to most analyses in the literature. When one looks at figure 1 and sees no response in taste-tasteless it is misleading because it seems as though the study was underpowered or something was wrong. I think many readers will come to this conclusion as the text no stands and therefore lose confidence in the other findings.

Demonstrating taste type coding even after controlling for palatability is a key component of our findings. We believe this is the most novel contribution to the literature, for which our findings suggest MVPA (Figure 2) over univariate techniques (Figure 1) for future research that uses fMRI. While we understand the reviewer’s concerns about including a positive taste-tasteless image in Figure 1, we felt it would undermine this key point. We also considered the alternative of including both controlled

and uncontrolled palatability images in Figure 1, but felt this would add confusion to the figure rather than emphasizing what is novel in the manuscript.

Instead, we felt the best compromise is to emphasize Figure S2 in the text, which as the reviewer asks shows the significant maps not controlled for palatability image congruent with previous meta analyses (lines 106-110).

Pg 7 (lines 106-110).

“As an important confirmation of prior findings¹⁵, univariate analyses not controlling for valence revealed consistent neural correlates of bitter taste experience relative to tasteless in the whole brain, activating the insula, frontal operculum, parietal operculum, thalamus and cingulate gyrus (Fig S2 and Table S2; conducted at a more liberal threshold $FDR < 5\%$)”

REVIEWERS' COMMENTS:

Reviewer #2 (Remarks to the Author):

The authors have adequately addressed my remaining concerns.